# Combined Magnetic Hyperthermia and Photothermia with Polyelectrolyte/Gold-Coated Magnetic Nanorods

**DOI:** 10.3390/polym14224913

**Published:** 2022-11-14

**Authors:** Marina Lázaro, Pablo Lupiáñez, José L. Arias, María P. Carrasco-Jiménez, Ángel V. Delgado, Guillermo R. Iglesias

**Affiliations:** 1Department of Applied Physics and Instituto de Investigación Biosanitaria ibs.GRANADA, NanoMag Laboratory, University of Granada, 18071 Granada, Spain; 2Department of Applied Physics, NanoMag Laboratory University of Granada, 18071 Granada, Spain; 3Department of Pharmacy and Pharmaceutical Technology, University of Granada, 18071 Granada, Spain; 4Department of Biochemistry and Molecular Biology I, Faculty of Sciences, University of Granada, 18071 Granada, Spain

**Keywords:** gold coating, hyperthermia, magnetic nanoparticles, nanorods, photothermia, polyelectrolyte layer, polyethyleneimine

## Abstract

Magnetite nanorods (MNRs) are synthesized based on the use of hematite nanoparticles of the desired geometry and dimensions as templates. The nanorods are shown to be highly monodisperse, with a 5:1 axial ratio, and with a 275 nm long semiaxis. The MNRs are intended to be employed as magnetic hyperthermia and photothermia agents, and as drug vehicles. To achieve a better control of their photothermia response, the particles are coated with a layer of gold, after applying a branched polyethyleneimine (PEI, 2 kDa molecular weight) shell. Magnetic hyperthermia is performed by application of alternating magnetic fields with frequencies in the range 118–210 kHz and amplitudes up to 22 kA/m. Photothermia is carried out by subjecting the particles to a near-infrared (850 nm) laser, and three monochromatic lasers in the visible spectrum with wavelengths 480 nm, 505 nm, and 638 nm. Best results are obtained with the 505 nm laser, because of the proximity between this wavelength and that of the plasmon resonance. A so-called dual therapy is also tested, and the heating of the samples is found to be faster than with either method separately, so the strengths of the individual fields can be reduced. Due to toxicity concerns with PEI coatings, viability of human hepatoblastoma HepG2 cells was tested after contact with nanorod suspensions up to 500 µg/mL in concentration. It was found that the cell viability was indistinguishable from control systems, so the particles can be considered non-cytotoxic in vitro. Finally, the release of the antitumor drug doxorubicin is investigated for the first time in the presence of the two external fields, and of their combination, with a clear improvement in the rate of drug release in the latter case.

## 1. Introduction

Magnetic nanoparticles (MNPs) are widely used in many different fields, but their potential in biomedicine stands out, particularly in the investigation of cancer treatment and diagnosis [1,2,3]. Although some are based on cobalt, nickel, or zinc and thus have moderate biotoxicity, those based on iron oxides (maghemite, magnetite) are recognized as applicable nanomaterials for theranostic purposes. The large variety of particles that have been synthesized opens a varied range of applications, going from the possibility of being used as contrast media in MRI [4], to that of becoming effective, directed, and specific drug transport vehicles [5]. Their response to magnetic fields capable of directing them to specified parts of the body and keeping them there while carrying out their therapeutic role is another feature explaining their significance in the area. Furthermore, and this is the core of the present contribution, if the applied magnetic field is alternating, another application shows up, in which a lot of confidence has been deposited for at least two decades. This is the possibility of generating localized heat leading to so-called magnetic hyperthermia, which has become an innovative treatment, considered a potential adjuvant to other cancer therapies. The application of such fields (of proper frequency and amplitude) produces energy loss in the form of heat which can be localized in the tumor, leading to cancer cell death. In fact, in magnetic hyperthermia, if MNPs are suspended in an aqueous medium, it is possible to raise their temperature by applying AC fields with frequencies of 100–200 kHz and intensities around 20 kA/m, although results have been shown whereby frequencies as high as 600 kHz and field strengths up to 59 kA/m have been applied [6,7].

Another source of local heating has been devised in which MNPs (eventually modified) are subjected to the action of a light beam of suitable wavelength and intensity. By different mechanisms, particles can absorb part of that (visible or IR) radiation and re-emit it in the form of heat. The method is called photothermia, and, interestingly, both electromagnetic stimuli (AC magnetic fields or light irradiation) can be used separately or together to improve the sought response [8]. Although photothermia has been tested with different materials [9,10,11], we focus on the use of the same magnetic particles for both heating techniques.

The basis of both therapies is the fact that tumor cells do not survive temperatures above 41–46 °C, even for a short period of time [12]. It is well known that such cells have accelerated metabolism, as well as a disorganized vascular structure, that is, they are highly vascularized but with poor-quality vessels. Therefore, when they are subjected to heating, the blood flow is not efficient enough to dissipate the heat and it is possible to cause apoptosis by temperature elevation without significantly affecting the surrounding healthy tissue. Moreover, it is difficult to eliminate nanometric objects that enter the tumor cell, which will allow them to spend more time inside it. This effect is therefore called enhanced permeability and retention (EPR [13]). In addition, it should also be noted that the pores of the vessels determine the size of the particles that can be extravasated: the windows are of the order of, at most, 300–700 nm [14].

The amount of heat produced is evaluated by the quantity called the specific absorption rate (SAR, or amount of heat released per unit mass of magnetic material) parameter. When MNPs are exposed to an electromagnetic stimulus, the SAR is affected by extrinsic factors such as the frequency and strength of the applied field, as well as the particles’ intrinsic properties such as stability, saturation magnetization, anisotropy energy, and rate of magnetic relaxation [15]. It is usual to also define the intrinsic loss power (ILP, the ratio of SAR to the product of the squared field strength and the frequency), which normalizes the heating rate values and makes them almost independent of the frequency and strength of the applied field, thus allowing comparison between different devices, particles, and laboratories [16].

Size determines the optimum frequency for magnetic hyperthermia application, and it is also an essential factor for the stability of the MNP dispersion. Morphology is also important; most works with biomedical focus have been performed with spherical magnetite particles, but an increasing number of contributions demonstrate the advantage of nanorods, particularly in drug transport and release. Truong et al. [17] have recently reviewed the field, suggesting exciting possibilities in the use of nanorods, mainly considering the flexibility in sizes and axial ratios that can be achieved, confirmed by works demonstrating that the tumor cells can capture elongated particles with short dimensions very efficiently, without increased biotoxicity [18,19,20]. Some drawbacks must also be mentioned: in the case of very elongated particles, the only route for membrane penetration is mechanical disruption, because of hindered endocytosis [20]. However, it may be worth pointing out that such mechanical penetration may be used with advantage for tumor cell destruction if rotating magnetic fields are applied [21,22,23].

Several studies have thus concentrated on anisotropic forms to improve the magnetic behavior of the particles (increased anisotropy related to shape is beneficial for our heating objectives) and drug delivery [24,25,26]. The most frequently used materials are magnetite (Fe_3_O_4_) and maghemite (γ− Fe_2_O_3_) since they are magnetic particles accepted for clinical use in Europe and the USA. This is because of their strong magnetization and minimal toxicity, the latter being greatly improved by proper surface treatments of the particles [27]. In fact, although the magnetic core plays a fundamental role in the pursued applications, the easy aggregation and degradation of iron oxides in biological fluids prevent the use of the bare particles in biomedical uses. Ideally, particles should have diameters below 6 nm in order to be excreted through the kidney [28]. However, few syntheses produce such small particles, and their use by injection and transport by an externally applied magnet would be difficult due to their Brownian random motions. Whatever the size, untreated particles are prone to adsorb plasma proteins (opsonins) that make them targets for macrophages of the mononuclear phagocyte system, leading to changes in the particles themselves and in the pH of the medium, affecting cell viability [29,30,31]. Coating with inert or biocompatible shells, including polymers, silica, or inert metals such as gold or silver [28,31,32,33,34,35], appears necessary for reducing the potential toxicity of the particles.

Many tests of cell toxicity are performed in vivo, clearly simpler, and more accessible than in vivo evaluations, but this brings about the issue of the correspondence between both types of results, with poor correlation found in most cases. Mahmoudi et al. [30] analyzed the possible routes to increase the applicability of in vitro toxicity evaluations as a predictor of in vivo results. Focusing on the particles of interest in this work, Maniglio et al. [36] recently demonstrated that magnetite/gold composite nanoparticles 12 nm in diameter did not show any significant toxicity against MG63 and NIH/3T3 cell lines, confirming results from other authors [37,38] in relation to the negligible cytotoxicity of gold-coated magnetite. A one-year study performed by Kolojsnaj-Tabi and coworkers [39] disclosed many aspects of the fate of gold/magnetite nanocomposites in vivo, after injection in laboratory mice. It is worth noting that the degradation of magnetite may enter the route of iron metabolism by cells and be finally managed by the body of the living animal without provoking further harm, although at the price of losing the original magnetic properties. These authors found that particles accumulated in lysosomes of Kupffer cells in liver and macrophages in spleen. After long-term observations, gold particles were found to remain rather stable (with some reduction in diameter), whereas only traces of iron oxide were appreciable. This was hence eliminated by dissolution, and not as solid nanoparticles.

Coating with gold has an additional advantage. Although magnetite NPs by themselves have a photothermal response to a λ ≈ 800 nm laser excitation, as demonstrated by Espinosa et al. [40] pursuing multifunctional uses of these nanosystems, the combination of MNPs with metallic surfaces, especially gold layers, has gained importance in recent years. This is due to the possibility of adding an extra optical response to the magnetic one and, hence, maximizing the power absorption when exposed to an electromagnetic field. It should be noted that in vivo experiments are already being carried out with these types of techniques. One relevant example of a combination of physical therapies is a study based on patients with glioblastoma multiforme, which presented relevant results using magnetic hyperthermia in combination with radiotherapy [41].

It is worth mentioning that it has recently been shown that some ferrites may show photoluminescent behavior: the substitution of Fe^2+^ by Mn^2+^ in magnetite or in cobalt ferrite gives rise to such optical activity [42,43]. Interestingly, this may open an additional functionality for the MNPs as sensors and active devices. Thus, Ortgies et al. [44] make use of hybrid structures (MNPs and infrared emitting PbS quantum dots) to track, and thus observe, deep tissue images with higher penetration by magnetic resonance and luminescence. However, the role of such structural changes in the photothermal response can also be a field of application, yet unexplored.

The attachment of gold nanoparticles to magnetite has been investigated by different authors, and the stability of the coating and its uniformity and thickness are always an issue to consider. In the present approach, the core particles were coated with a cationic polyelectrolyte, poly(ethylene imine) (PEI), in order to promote the adhesion of negatively charged gold nanoparticles, as well as increasing the stability of the composite particles against aggregation [45,46]. This improved stability has been taken advantage of in magnetic hyperthermia, using different stabilizing routes [47,48].

The use of this polymer in the biomedical field has been strengthened by its application in gene delivery as a therapy in the treatment of different diseases due to its ability to electrostatically bind DNA [49], an alternative to viral vectors presently showing the best performance in this task [50,51]. There are, however, concerns about the cytotoxicity of PEI [49,52,53]. For instance, Hu et al. [54] found that cardiovascular toxicity in zebra fish embryos was associated with branched PEI with a molecular weight of 25 kD, and that toxicity increased with molecular weight. Although this is the most usual vehicle of DNA for gene transfection, lower molecular weights appear as less toxic to cells, with a safe limit around 2 kDa [55,56], at the price of reducing transfection efficiency.

Interestingly, Wang et al. [57] found that building of a layer-by-layer assembly of citric acid and PEI significantly reduced the cytotoxicity of the polymer. This is an argument in favor of coating the gold particles attached to the magnetite cores investigated in this work with citrate anions to make them negatively charged. Another advantage of attaching gold to the particles was demonstrated by Arsianti et al. [29], who found that the cellular viability in the presence of magnetite/gold/PEI used as DNA vehicles was significantly higher than that measured in the absence of the gold layer.

In this work, we present the synthesis and characterization of magnetic nanorods (MNRs). Thanks to the use of highly controllable hematite templates, the particles obtained are very homogeneous in size and shape, a feature that hydrothermal methods often used cannot guarantee. They are coated with a triple polymeric layer, and the cytotoxicity of the nanorods for the HepG2 cell line is evaluated. Furthermore, gold seed nanoparticles are also adsorbed as a final layer, to attempt at improving the optical response of the MNRs when irradiated with less penetrating wavelengths, specifically those close to the surface plasmon. The composite particles are subjected to magnetic hyperthermia, photothermia and both techniques simultaneously applied, in a sort of dual therapy, never attempted with this kind of particle. In addition, their doxorubicin release rates are evaluated, also for the first time in the presence of the two external agents simultaneously applied.

## 2. Materials and Methods

### 2.1. Materials

All reactants used were commercially available: iron (III) chloride hexahydrate, potassium dihydrogen phosphate, sodium citrate tribasic dihydrate, sodium borohydride, branched polyethyleneimine (PEI, Mw ≈ 2000 g/mol), poly(styrenesulfonate) (PSS, Mw ≈ 7 × 10^4^ g/mol), and doxorubicin hydrochloride (DOX) were purchased from Sigma-Aldrich (Darmstadt, Germany). Absolute ethanol was provided by Scharlau, Germany. The water used was deionized and filtered through a 0.2 μm filter in Milli-Q Academic equipment (Millipore, Molsheim, France).

### 2.2. Methods

#### 2.2.1. Synthesis of Magnetite Nanorods

The synthesis of magnetite nanorods (MNRs) was carried out in two stages. First, hematite rods were obtained, which were subsequently transformed by reduction at high temperatures into magnetite. For the preparation of the hematite templates, 0.02 mol of FeCl_3_.6H_2_O was mixed with 4.5 × 10^−4^ mol of KH_2_PO_4_ and dissolved in 1 L of Milli-Q water [58]. The solution was kept at 100 °C in a Pyrex bottle sealed with Teflon for 6 days. Thereafter, the hematite suspensions were cleaned by centrifugation at 21,000 rpm for 15 min, and the supernatant was discarded and replaced by water. This process was repeated 3 times. Finally, the sample was redispersed in water, and dried overnight at 50 °C.

Lastly, it was necessary to transform hematite nanoparticles into magnetite. This process was carried out by reduction at high temperatures in a H_2_ atmosphere [59]. For this purpose, a tube furnace (Hobersal, Barcelona, Spain) was set at 350 °C while passing nitrogen-carrying ethanol. The N_2_ flow was bubbled in pure ethanol and passed along the tube at a rate of 3.83 L/h. At the selected temperature, the ethanol was oxidized, releasing hydrogen, which, in turn, reduced the hematite powder. The powder was introduced once the temperature was reached and kept in the ethanol/N_2_ atmosphere for 6 h. It is important to note that for this flow to scan the largest area of NPs possible, a glass tube with a flat and porous base (fritted glass) was used as sample container inside the tube.

#### 2.2.2. Preparation of Polymer-Coated Nanoparticles

In order to improve MNP polymer coating, the layer-by-layer method was used, as nanoparticles were coated with 3 successive layers of polyelectrolyte (cationic–anionic–cationic) [47]. The procedure involved the addition of 5 mL of MNP suspension (with 1 g/L particle concentration) dropwise into 5 mL of PEI (2 g/L) under ultrasonic agitation. The obtained suspension was kept for 90 min in an ultrasound bath and then it was left undisturbed for 30 min. The final step consisted of cleaning several times by magnetic decantation and redispersion in 5 mL Milli-Q water, maintaining the initial concentration. This procedure was repeated for the successive PSS and PEI coatings.

#### 2.2.3. Gold Shell on Polymer-Coated Nanoparticles

Gold seeds were synthesized by adding 0.5 mL HAuCl_4_ 0.01 M and 0.5 mL sodium citrate 0.01 M to 18 mL of Milli-Q water under stirring. Afterwards, 0.5 mL of freshly prepared (and kept at 0 °C) NaBH_4_ 0.1 M was added. Sodium borohydride acts as a reducing agent, causing the reduction of Au^3+^ to neutral gold atoms, changing the solution from a yellowish to reddish color. The mixture was kept at 30 °C in a thermostatic bath for 15 min and left unperturbed for 2 h [47].

The procedure to attach the gold nanoparticles to the surface of the PEI/PSS/PEI-coated MNRs started by diluting the starting magnetite and gold suspensions. This was carried out by adding 20 mL of water up to a final concentration of 0.25 mg/mL, and they were added dropwise to a dilution of 30 mL of gold seeds and 10 mL of water, in an ultrasound bath. The resulting solution was maintained under sonication for 15 min and left unperturbed for half an hour. Finally, the particles were cleaned by magnetic decantation and redispersed in 2 mL of water, reaching a concentration of 2.5 mg/mL.

#### 2.2.4. HRTEM Characterization: Morphology

The size and shape of dried nanoparticles were determined using a high-resolution transmission electron microscope (HRTEM) (Thermo Fisher Scientific TALOS F200X, USA), all images being provided by the CIC-UGR Microscopy Service. The sizes were measured using JImage software (University of Wisconsin, Madison, WI, USA).

#### 2.2.5. Electrophoretic Mobility and Hydrodynamic Diameter Measurements

The determination of the surface electrical properties was carried out by measuring the electrophoretic mobility of dilute MNP suspensions using a Zetasizer Nano-ZS (from Malvern Instruments, Worcestershire, UK). The same instrument was used for the measurement of particle hydrodynamic diameters by dynamic light scattering (DLS).

#### 2.2.6. Structural Characterization

Crystal structures were identified from X-ray powder diffraction (XRD) patterns of the samples using a Bruker D8 Advance diffractometer (Berlin, Germany) equipped with a Bruker LINXEYE detector and a CuKα radiation source. Micro-Raman spectroscopy was performed using a JASCO NRS-5100 Dispersive Micro-Raman (Tokyo, Japan) equipped with a green diode laser of 532 nm, with 30 mW power (Elforlight G4-30; Nd: YAG), with a spectral range from 50 to 8000 cm^−1^. Thermogravimetric analysis was carried out in a Shimadzu TGA-50H (Kyoto, Japan) with a vertical furnace design, a maximum precision of 0.001 mg, and a temperature range between 25 °C and 950 °C (CIC-UGR).

#### 2.2.7. Magnetic Properties

Magnetization cycles were obtained at room temperature (20 °C) in an AC magnetometer AC Hyster Series (Madrid, Spain). It works at a frequency of 1 kHz, and it can apply field strengths in the range −1.5 to +1.5 kOe (−119.4 to +119.4 kA/m).

#### 2.2.8. Cytotoxicity Determinations of PEI/PSS/PEI-Coated MNRs

The human hepatoblastoma cell line HepG2, supplied by the European Collection of Animal Cell Cultures (Salisbury, UK), was used to perform cell viability experiments in order to prove the negligible toxicity of the triple polymer layer-coated MNPs. The cells were seeded into 96-well plates (10 000 cells/well) and cultured in a cell culture medium for 24 h with different concentrations of polymer-coated MNRs (25, 50, 100, 200, 300, 500 μg/mL) in a CO_2_ (5%) atmosphere. After the treatment was completed, the medium was removed from each well by aspiration, and the cells were fixed with 100 μL of glutaraldehyde. The plate was stirred for 15 min at 50 rpm and it was then washed with distilled water 8 times. After drying, 200 µL of 0.1% crystal violet was added and stirred for 20 min at 30 rpm. Again, the plate was washed and dried. Finally, the dye was solubilized with 100 μL of 10% acetic acid. The plates were stirred for 10 min and introduced into a plate reader set at 590 nm wavelength, which provided the results of live cells [60].

#### 2.2.9. Optical Absorbance

UV/Vis optical absorbance of all samples was determined using a Jenway Series 67 (UK) spectrophotometer.

#### 2.2.10. Magnetic Hyperthermia

The magnetic hyperthermia device was built in our laboratory. The AC magnetic field was applied to the samples using a double four-turn coil 32.40 ± 0.05 mm in diameter, made of 6.00 ± 0.05 mm copper tube. An AC current of frequency *f* was passed along the coils using a parallel LC resonant circuit. The inductance was that of the coil (*L* = 2.48 ± 0.01 mH, *R* = 300 ± 1 mΩ), and the device contained a capacitor bank for selecting the frequency in the range 100–300 kHz. In the present work, the coil remained unchanged, and only the capacitance of the circuit capacitors was used to select the frequency of the AC magnetic field. The copper tube of the coil allowed circulating water from a thermostatic bath. This is important, since the Joule heating can be very significant (currents as high as 40 A are passed through the coil) and could easily hide any temperature changes associated with true magnetic hyperthermia. The resulting field strength can be 22 kA/m at most. Field strength and frequency measurements at the sample location were determined by a NanoScience Laboratories Ltd. Probe (Staffordshire, UK), with 10 μT instrumental sensitivity.

An Eppendorf tube containing the sample (200 μL) was placed centered inside the coil, after being thermostated at an initial temperature of 20 °C. Sample and bath temperatures were registered using two optical fiber sensors (Optocon AG, Dresden, Germany) every 3 s with 0.01 °C resolution.

#### 2.2.11. Photothermia Application

The setup consisted of an infrared laser (Laserland, Dresden, Germany, 850 nm, maximum power 1.6 W) pointing towards the support where the Eppendorf was located with the sample (200 μL). A thermal imaging camera (Flir 60, Niceville, FL, USA) capable of taking temperature measurements every 0.0625 s with 0.045 °C thermal resolution was used for sensing the sample temperature. To compare with the results obtained from the determination of the optical absorbance, less powerful red, green, and blue lasers with respective wavelengths of 638 nm, 505 nm, and 480 nm (RGB Combined White Laser, Laserland, Dresden, Germany) were also used.

#### 2.2.12. Dual Magnetic Hyperthermia and Photothermia Device

When magnetic field and laser light were applied simultaneously, the laser was directed towards the top of the uncapped Eppendorf containing the sample. In this case, only the thermographic camera could be used for temperature measurements. Infrared laser irradiation and a constant-frequency (100 kHz) field were used in this combined setup.

#### 2.2.13. Drug Loading and Release

For the first step, drug loading on the MNRs, 1 mg of gold-coated nanoparticles was kept in contact with 1 mL doxorubicin hydrochloride (DOX) solutions of different concentrations and maintained for 72 h under shaking. In order to determine the DOX concentration after adsorption, a calibration set of absorbance vs. concentration data was obtained by measuring optical absorbance at 483 nm (wavelength of maximum absorbance for DOX) for the concentrations used (0.02, 0.04, 0.06, 0.08, 0.10, and 0.12 mM DOX).

The second step was drug release experiments. For this purpose, 1 mL of 5 mg/mL suspension of gold-coated MNPs was placed in 1 mL 0.6 mM DOX and kept under shaking for 72 h. After this time, the particles were magnetically decanted, and the DOX solution was substituted by acetate buffer at pH 5, a value selected considering that the tumor tissue environment is acid [61]. At specified intervals, the particles were decanted, and the absorbance of the supernatant measured to evaluate the amount of released DOX, using the calibration data. Fresh buffer was added, and the release process was continued (sink conditions [47]). Release experiments were carried out in four different experimental conditions, namely, under magnetic hyperthermia, in photothermia, with dual therapy, and in standard conditions (no field applied).

## 3. Results and Discussion

### 3.1. Size and Shape Characterization

Figure 1a,b show HRTEM images of the MNRs synthesized. They clearly demonstrate their geometry, spheroidal, as determined by the hematite template NPs. In order to carry out the geometric characterization, the length of at least 100 particles was determined in different pictures, and the value obtained for the average length was 550 ± 90 nm, with an aspect ratio of 1:5 (a detailed size distribution can be found in the Appendix A).

Concerning the gold seeds, HRTEM pictures showed mostly spherical particles, quite monodisperse (Figure 1c,d). Their average diameter was estimated by DLS, obtaining a value of 6.00 ± 0.10 nm. In addition, Figure 1c,d demonstrate that a uniform coating of the magnetic nanorods was achieved, most likely a consequence of the PEI/PSS/PEI layer on magnetite, which enabled electrostatic bonds between the MNPs and the gold particles. It is worth noting that the gold shell is not continuous but particles are deposited individually or form gold islands. This has been found by other authors when coating magnetic particles with gold [47,62] or silver [63], and calculations based on the density functional theory, performed by Sokolov et al. [32], demonstrated recently that the wetting parameter of the magnetite/gold interface happens to be negative, so that gold does not wet magnetite and it rather forms drops on its surface. In addition to morphological evaluation of the gold shell, EDX composition maps of the bare and gold-coated particles were also obtained, as shown in Figure 1e,f for bare and gold-coated MNRs.

### 3.2. Electrophoretic Measurements

Measurements of electrophoretic mobility as a function of pH served as a qualitative assessment of the surface structure of the NPs. The use of branched, low-MW PEI has initially been justified as a biocompatible polymer. However, one of the questions we can ask ourselves is whether the polyelectrolyte layer would detach over time, i.e., whether the particles might not be stable and gradually increase their cytotoxicity by release of free polymer, since it has been shown that a single layer can detach partially as time passes [47]. To verify if this was the case with our particles, the electrophoretic mobility of magnetite/PEI particles was measured first right after their preparation, finding that it was positive throughout the pH range. However, when the same measurements were taken a few days later, the loss of positive charge was evident, indicating detaching of the PEI layer. That is why the particles were coated with a triple layer using the layer-by-layer technique, in the order PEI (cationic)/PSS (anionic)/PEI (cationic) leading to an increased stability over time. By following this procedure, this third PEI layer remained attached to the particle, as is shown in Figure 2.

Coatings with the two polyelectrolytes changes the surface charge of the bare particles, as expected. The electrostatic bond between the layers allows the generation of this type of multilayered particle and finally their decoration with gold seeds through attraction between positive PEI and negative citrate-capped gold.

In general, in all the cases (bare, polymer- and gold-coated magnetite), the isoelectric point was found at acidic pH (around 5 for the gold-coated sample, and 6 for the multilayered sample). This finding is significant for particles aimed at DOX delivery, considering that this drug is positively charged at that pH range, and that the tumor environment is typically acidic [64,65,66]; this means that DOX will tend to be released from the particles not only by diffusion, but also by electrostatic repulsion. These mobility data can be found in Appendix A.

### 3.3. Structural Characterization

It can be seen in Figure 3a that a complete transformation of hematite into magnetite appears to be achieved in view of the XRD results (magnetite reference data taken from RRUFF database [67]). It can be also noted that the diffraction pattern is not modified by the coating with polyelectrolytes, and that when gold is added, its characteristic peaks, although weak (because of the small amount of gold, see Figure 1) are also appreciated. However, since the XRD diffraction pattern of magnetite and maghemite are very similar, other methods must be used to ascertain the structure. In addition to XPS, Raman spectroscopy is one possibility. The micro-Raman spectroscopy results obtained with our sample of MNRs are plotted in Figure 3b: the bands observed correspond [67] to magnetite (309, 560, and 690 cm^−1^, RRuff ID R060656.3) and maghemite (288, 406, and 495 cm^−1^ reference data, Rruff ID R140712), indicating that our particles are in fact a mixture of maghemite and magnetite, although the magnetite denomination will be used throughout the manuscript for simplicity.

### 3.4. Thermogravimetric Analysis

Thermogravimetric analysis (TGA) was performed to estimate the amount of polymer forming the triple layer around the particle. Figure 4 shows the evolution with temperature of the sample mass for both naked and PEI/PSS/PEI-coated nanorods. The bare particles do not show any special feature, except for a slight weight gain above 300 °C, which can be ascribed to partial oxidation of magnetite to hematite or maghemite [68]. In the case of the coated particles, the first significant weight change happens around 150 °C and it must correspond to the loss of hydration water, while the next event extends up to 367 °C. Since the decomposition temperature of PSS is around 470 °C [69], it can be argued that this stage was the loss of PSS. This loss is about 1.05%. The same TGA data allow us to observe a weight loss of 4.07% in a process that finishes around 605 °C. Literature information on PEI polymer indicates that it decomposes at around 600 °C [70] with the TGA curves showing a pronounced weight loss. The data in Figure 4 confirm the coating of the particles by the polyelectrolytes, especially PEI. An estimation of the thickness of the respective layers has been carried out by assuming that the density of such layers is at most 1/10 of that of the free polymers (1.1 g/cm^3^ and 1.08 g/cm^3^, respectively). If a uniform coating is admitted (as suggested by the pictures in Figure 1C and 1D), it can be estimated that the thickness of the PSS layer is 2.4 nm, and that of the PEI rises to about 10 nm (compare the layer appreciated in Figure 1C with the size of the Au NPs).

### 3.5. Magnetic Characterization

The magnetization cycles of MNRs, both bare and coated by polymer or polymer/gold, are displayed in Figure 5. As observed, the bare MNR sample showed a high saturation magnetization, typical of that of good magnetite samples, namely 71.2 ± 3.0 emu/g. The facts that the measurements were performed in AC magnetic fields, and that the particles are multidomain, produces open hysteresis cycles. As expected, a decrease in magnetization is observed with successive coatings (polymer-coated MNRs present a maximum magnetization of 21.3 ± 1.8 emu/g, and 12.1 ± 0.7 emu/g is found for gold-coated ones).

### 3.6. Cell Viability Experiments

Cell viability experiments were performed to determine whether the coatings provided the necessary biocompatibility of the systems. For this purpose, HepG2 cells were seeded for 24 h with different concentrations (25, 50, 100, 200, 300, 400, 500 μg/mL) of PEI-PSS-PEI-coated nanoparticles. The experiments were conducted in triplicate, and the viability, normalized with respect to the control cells, is evaluated as the mean ± SD of these experiments. As can be seen in Figure 6, the preliminary data are very promising. The cytotoxicity of polyelectrolyte-coated MNRs particles is negligible, even at the highest concentrations. No significant differences were observed between cells cultured with the concentration sweep and control cells (within statistical error).

### 3.7. Optical Absorbance

The results of the optical absorbance determinations are depicted in Appendix A: the shoulder shown by the absorbance of the gold-coated sample indicated the presence of gold, since such an absorbance maximum can be ascribed to the surface plasmon of gold particles, observed at 564 nm. This is another proof that an efficient coating was achieved.

### 3.8. Magnetic Hyperthermia Experiments

The magnetic hyperthermia performance of the different nanorods was evaluated for different frequencies and intensities of the AC magnetic field. The results are presented in Figure 7 (temperature vs. time) and Figure 8 (SAR and ILP). As can be seen from the SAR values, the best frequency for this application and this sample was 150 kHz. This means that the phase shift between magnetization and field strength is largest at this frequency, thus producing the maximum heat dissipation.

Magnetic hyperthermia experiments were also carried out with the gold-coated particles (Figure 9a). As expected, given the magnetization cycles described above, the efficiency with gold coating decreased, as confirmed by the significant reduction in SAR (Figure 9b).

### 3.9. Photothermia

In photothermia, two approaches were followed: the first, considering the presence of the surface plasmon of gold in the visible spectrum (Appendix A), consisted of irradiating with wavelengths close to it, using the RGB laser. Although penetration is lower, it may still be useful in certain cases of superficial cancer [71], and, in our case, as a proof of the improvement achieved by coating with gold NPs. The second was to irradiate the sample with a near-infrared (NIR) laser due to its great penetration through human skin.

Figure 10 shows the results of the first approach: note that for monochromatic laser the coating with Au improved the photothermia response, as compared to that of MNRs with the triple polymer shell. Taking into account that the laser powers were not identical for the three wavelengths tested, we could calculate the ratio SAR/(power density) for each of the lasers. The results were (in units of cm^2^/g) 0.21, 0.22, and 0.17 for 480, 505, and 638 nm, respectively, so that the 480–505 nm illumination source produced the best photothermia performance, as one would expect given the proximity between this wavelength and that of the surface plasmon resonance (SPR) of gold nanospheres.

In the case of NIR irradiation (Figure 11), it is observed that when the MNR particles are coated with gold NPs the rate of heating is reduced in comparison with that reached with polymer-coated MNRs, mainly at high laser power density. Considering that the plasmon resonance of our gold nanoparticles is around 570 nm (Appendix A), it can be explained that the presence of gold seeds does not help in heating by the SPR effect, but apparently reduces the heat transfer by radiation without actively participating in heating. In contrast, Rincón-Iglesias et al. [72] recently reported a significant photothermal heating with 140 nm magnetite rods coated by a continuous layer of gold, using a 0.2 W/cm^2^ NIR led source of 735 nm in wavelength. These experiments demonstrate that controlling the thickness and uniformity of the coating is an essential aspect of hyperthermia with composite magnetic nanorods.

### 3.10. Dual Therapy

As mentioned, by dual therapy we mean the situation in which an AC magnetic field is applied simultaneously to the photothermia laser, with the aim of reaching a given therapeutic response while decreasing the energy inputs from both the magnetic field (17 kA/m amplitude instead of 21 kA/m) and the laser (0.3 W/cm^2^ instead of 1 W/cm^2^). The combination can thus be thought of as a potentially less harmful, although equally efficient, treatment. The experimental data can be found in Appendix A.

This is in contrast with the results achieved when both therapies are superimposed. As Figure 12 shows, when the samples are irradiated with 0.3 W/cm^2^ while being subjected to a magnetic field of 17 kA/m, the rate of temperature increase is significantly faster, and as a result the SAR values reached with the dual therapy are three times larger than those corresponding to either technique separately. This is a very important result, as it opens a way of optimizing the treatment while minimizing unwanted size effects.

### 3.11. Drug Release

As repeatedly mentioned in this contribution, one possible application of the particles described is drug release alone or in combination with the hyperthermia effects. To this aim, polyelectrolyte–gold-coated MNRs were first contacted with a doxorubicin solution of 0.6 mM concentration, as described. Optical absorbance of the supernatant at the end of this time indicated a DOX concentration of 0.146 mM. This means that the adsorption was 0.308 μmol/0.005 g, equivalent to 61 *μ*mol per gram of magnetic particles or 33 mg/g. This adsorption density is of the same order of the values found in other works for adsorption of DOX on functionalized magnetic particles. Thus, Reyes Ortega et al. [73] reported 20 mg/g for spheroidal magnetite particles coated with PEI/PSS, and Rudzka et al. [74] obtained a saturation adsorption of 25 mg/g using particles with a maghemite/silica/gold composite structure. Using a different particle structure (carbon dots and magnetite on carbon nanohorns), Su et al. [75] measured up to a 66 mg/g DOX payload. Comparable values can be found in other works [76,77,78]. The mechanism of DOX adsorption on the investigated Au-coated MNRs must be mostly electrostatic. It is well known that the amino group in the DOX molecule is characterized by a p*K_a_* value of 9.93 [79], so that the molecule will be positively charged at acidic pH. The presence of a carbonyl group in the molecule means that the drug presents a low positive charge in neutral conditions. The fact that the adsorbed gold nanoparticles are negative leads to an electrostatic attraction between such particles and DOX as the predominant adsorption mechanism. Optical absorbance data reported by Rudzka et al. [74] demonstrate that the surface plasmon resonance of gold is quenched after contact with DOX in solution, a proof of the proposed mechanism of interaction, also confirmed by the data from other investigators [80,81].

In order to evaluate the release of this drug, absorbance measurements were performed under sink conditions at pH 5. Figure 13 shows the results in the form of accumulated drug release as a function of time during application of magnetic hyperthermia (17 kA/m, 100 kHz), photothermia (850 nm wavelength, 0.3 W/cm^2^), and both, as well as without treatment (the latter data are extended to longer times in Appendix A). Assuming the mechanisms described above, it is expected that at neutral pH the release is slow and very limited, due to the reduced tendency of the drug to be solubilized, and in fact most published works have focused on acidic media for passive release, as in the present work. In such conditions, DOX is strongly positively charged and prone to be solubilized and released to the solution by diffusion. In addition, Figure 2 shows that the coated MNRs become increasingly positive, leading to electrostatic repulsion between DOX and the substrate, easing the release. The main result in Figure 13 is the improvement in the rate of drug release when both fields are applied simultaneously. In this case, almost 10% of the DOX available was released in less than one hour. Furthermore, this rate can be easily tuned by controlling the field parameters.

## 4. Conclusions

Magnetite/maghemite nanorods 550 nm long have been successfully synthesized based on hematite templates, and they have been coated with a triple PEI/PSS/PEI polyelectrolyte layer, allowing the final adsorption of gold nanoparticles. The particles were designed to be applied under alternating magnetic fields and produce magnetic hyperthermia as an adjuvant therapy against malignant cells. They are also demonstrated to be very efficient heating sources when subjected to laser beams (photothermia) both in the near infrared and in the visible spectrum, the former being preferred because of its better penetration in the body, and the latter because of proximity with the wavelength of the surface plasmon resonance of gold nanoparticles. The nanorods coated by polymer respond very efficiently to either of the external (magnetic or electromagnetic) actions, whereas the coating by gold was only beneficial if the laser was in the visible window of the spectrum. In this work, we also describe a sort of *dual therapy*, consisting of the simultaneous application of a magnetic field and an infrared laser. The temperature window in which cancer cells are supposed to undergo destruction is reached with this kind of therapy in a shorter time and with lower (safer) strengths of the magnetic field or the laser power. The gold-coated magnetite nanorods are also used as drug vehicles for doxorubicin, and it is also shown that dual therapy provides better results for the drug release. The particles could be a multitask tool in fighting cancer cells. In future works, their use for cell hyperthermia in vitro, and the possibility of using them for mechanical disruption of the cells by applying rotating magnetic fields, will also be explored.

## Figures and Tables

**Figure 1 polymers-14-04913-f001:**
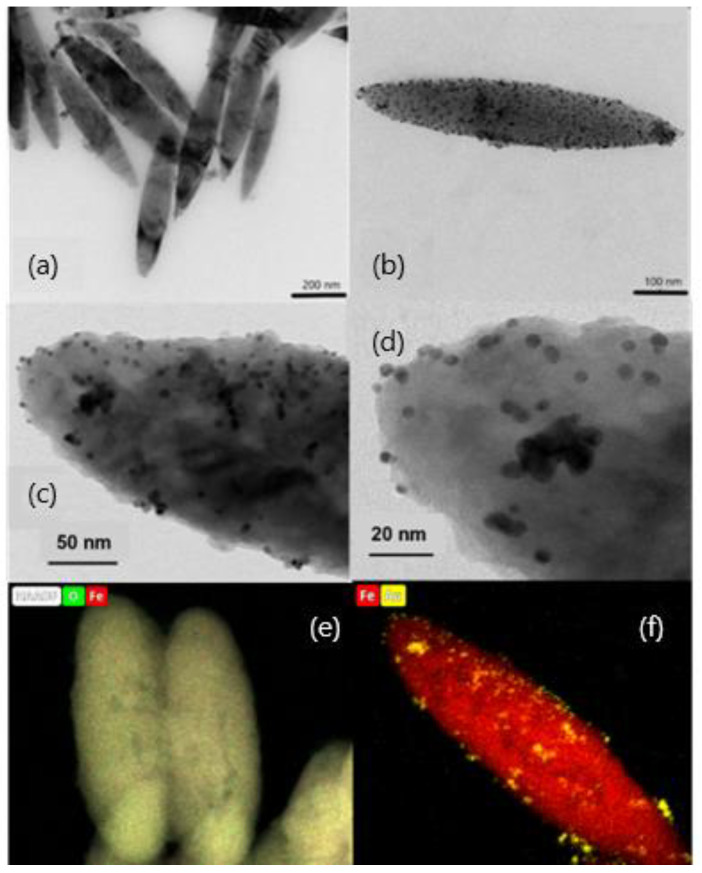
HRTEM images of magnetite nanorods; (**a**) bare particles; (**b**) gold-coated particles (the darker dots are Au NPs incorporated into the polymer shell); (**c**,**d**) high magnification of gold particle-coated nanorods, with observation of the polymer layer with gold embedded in it; (**e**,**f**) EDX composition maps of bare (iron and oxygen) and gold-coated (iron and gold) nanorods.

**Figure 2 polymers-14-04913-f002:**
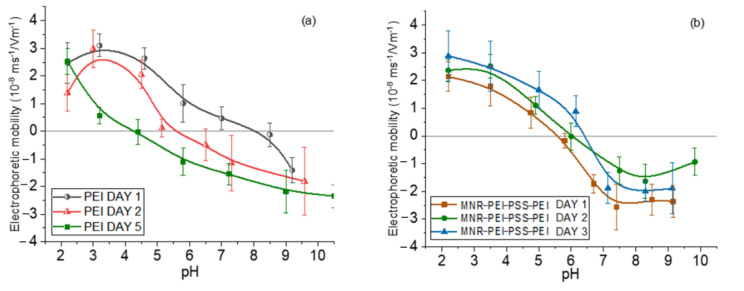
Electrophoretic mobility of magnetite nanorods at different aging times with one (**a**) and three (**b**) polyelectrolyte layers.

**Figure 3 polymers-14-04913-f003:**
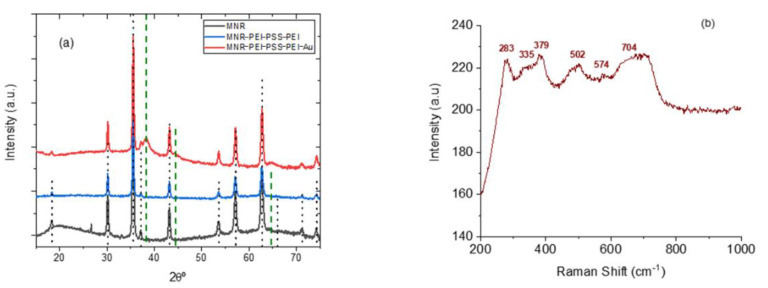
(**a**): X-ray diffraction patterns of bare, polymer-coated, and polymer/gold-coated MNRs; (**b**) micro-Raman spectroscopy of the bare particles. The vertical lines in (**a**) are the reference lines for magnetite [33]; dotted black lines: magnetite; dashed green lines: gold.

**Figure 4 polymers-14-04913-f004:**
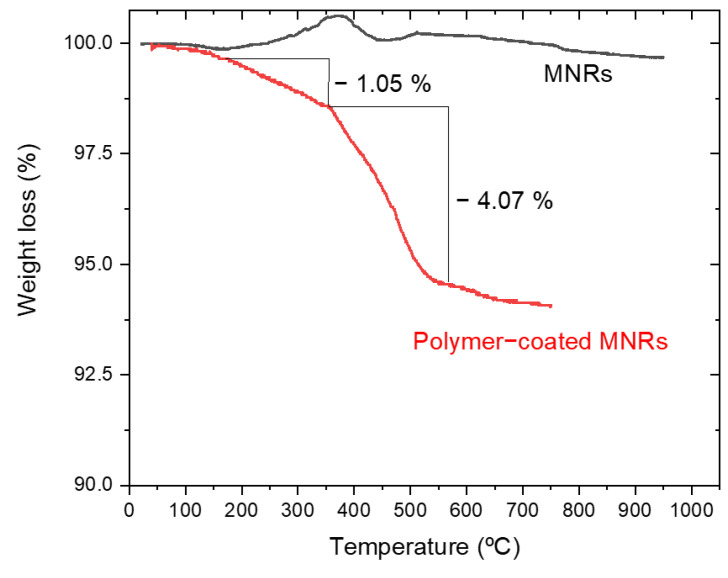
Thermogravimetric analysis of naked and polymer-coated MNRs.

**Figure 5 polymers-14-04913-f005:**
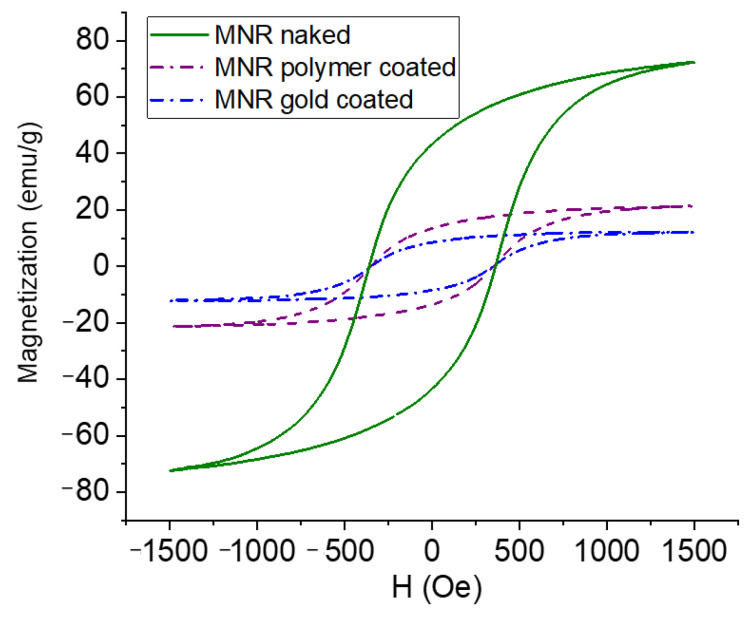
Magnetization curves of naked, polymer-, and polymer–gold-coated magnetite nanoparticles.

**Figure 6 polymers-14-04913-f006:**
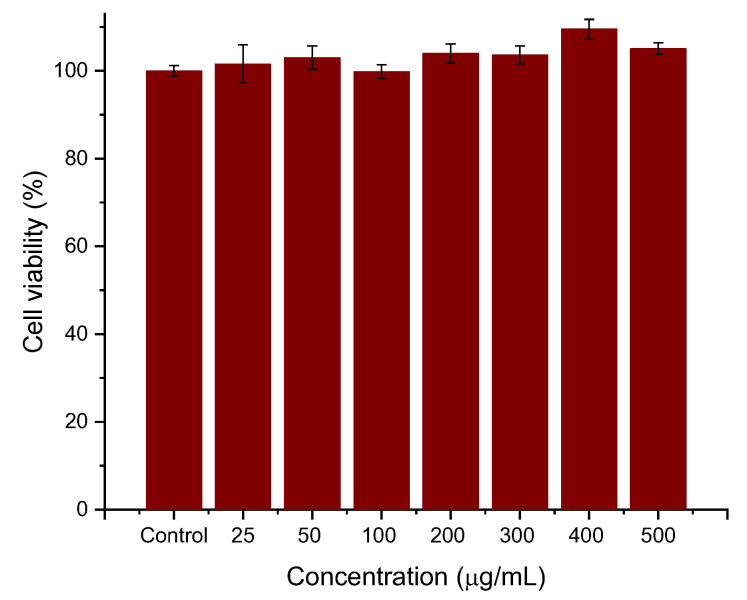
Effects of PEI-PSS-PEI-coated MNRs on HepG2 cell proliferation. The concentration of nanorods as indicated.

**Figure 7 polymers-14-04913-f007:**
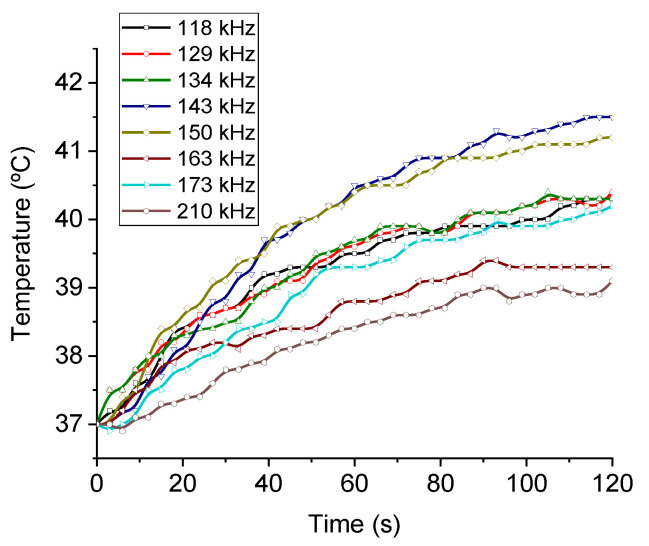
Frequency sweep for the MNR sample. Field intensity maintained at 17 kA/m.

**Figure 8 polymers-14-04913-f008:**
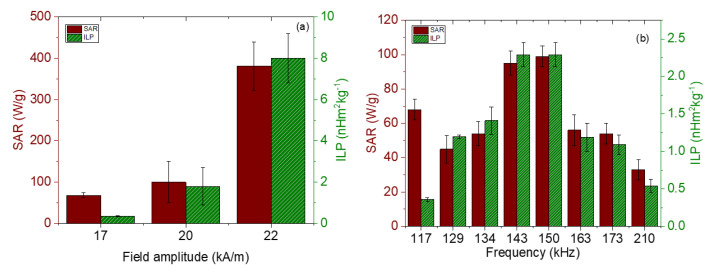
SAR and ILP values for magnetite nanorods. (**a**) Field strength sweep keeping the frequency at 100 kHz; (**b**) frequency sweep for a 17 kA/m field amplitude.

**Figure 9 polymers-14-04913-f009:**
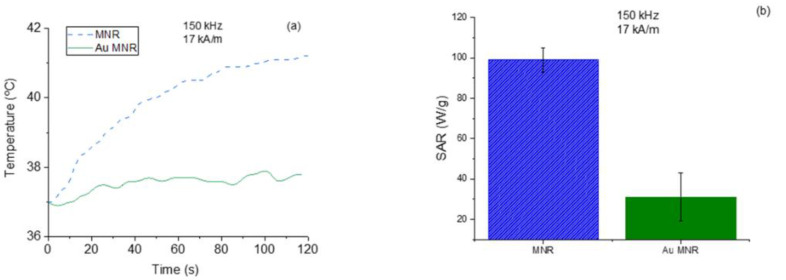
Temperature increases for polymer-coated and gold-coated MNRs (**a**), and SAR values (**b**).

**Figure 10 polymers-14-04913-f010:**
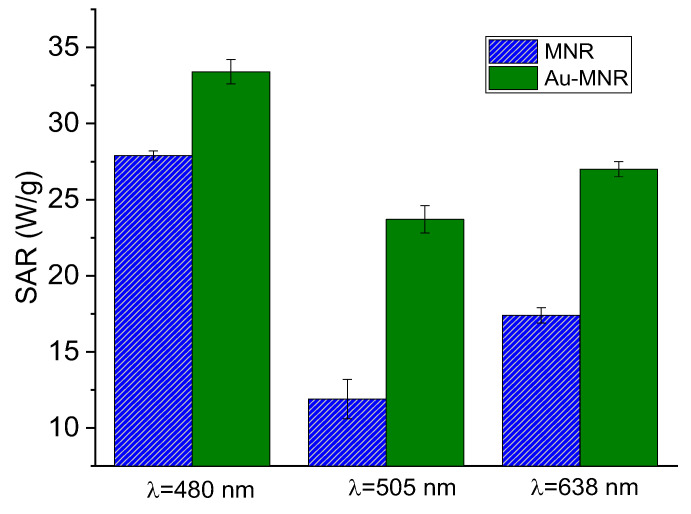
SAR values for RGB laser irradiation of polymer- and gold-coated magnetite samples. The respective laser power densities were 159 mW/cm^2^ (480 nm), 109 mW/cm^2^ (505 nm), and 161 mW/cm^2^ (638 nm).

**Figure 11 polymers-14-04913-f011:**
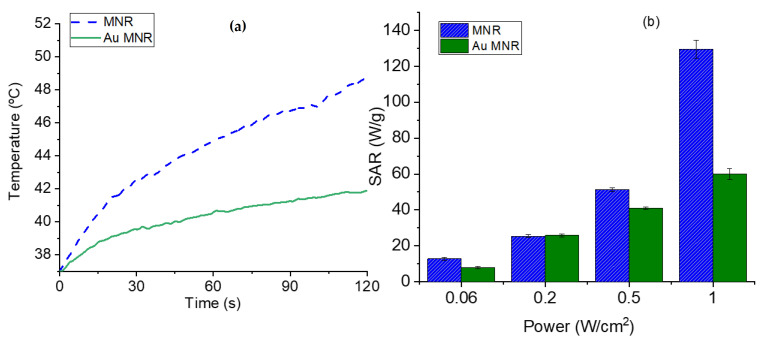
(**a**) Temperature evolution as a function of time in photothermia experiments (NIR laser, 850 nm wavelength, and 1 W/cm^2^ power density in the sample location), for polymer-coated and gold-coated MNRs. (**b**) SAR values as a function of laser power density for the two samples.

**Figure 12 polymers-14-04913-f012:**
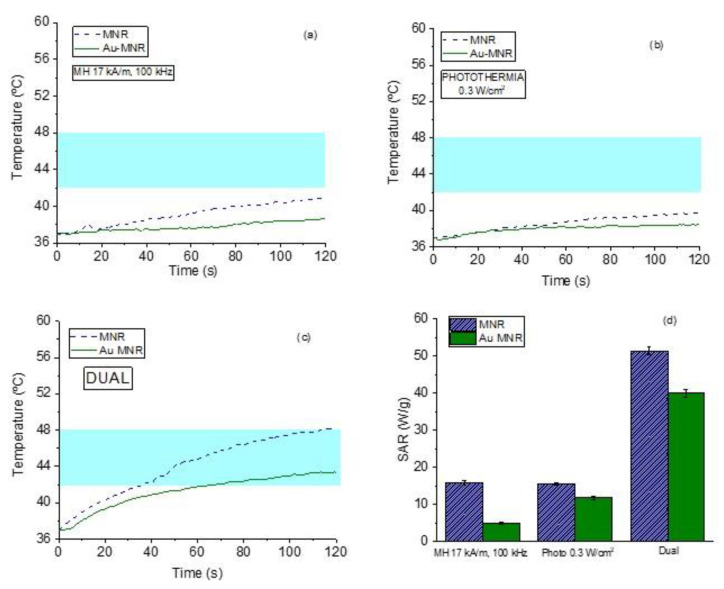
Magnetic hyperthermia (**a**), photothermia (**b**), and dual therapy (**c**) curves for MNR magnetite. (**d**) Comparison between SAR values of magnetic hyperthermia, photothermia, and dual techniques. The magnetic field frequency used is 100 kHz. The blue band marks the desired hyperthermia range.

**Figure 13 polymers-14-04913-f013:**
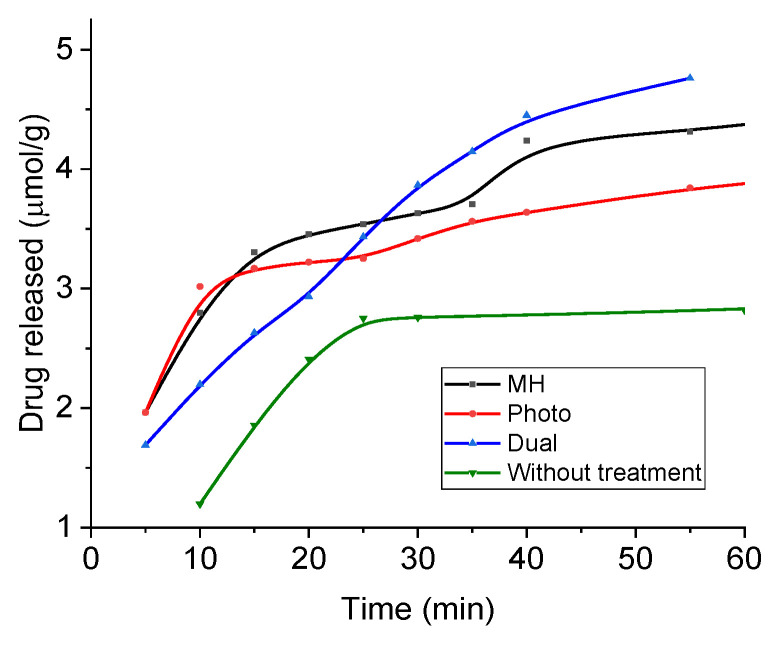
DOX release from magnetic nanoparticles in a pH 5 buffer and sink conditions. MH: under magnetic hyperthermia; Photo: with applied laser; Dual: both laser and magnetic field applied.

## Data Availability

All data pertaining to this contribution are contained in it.

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
