# Peer review of "Combined Magnetic Hyperthermia and Photothermia with Polyelectrolyte/Gold-Coated Magnetic Nanorods"

_polymers, 2022, doi:10.3390/polym14224913_

Round 1
Reviewer 1 Report
.
Author Response
COMMENT 1
- English language and style are fine/minor spell check required
- Must be improved
o English language and style are fine/minor spell check required. background and include all relevant references?
o Are all the cited references relevant to the research?
o Are the results clearly presented?
o Are the conclusions supported by the results?
- Can be improved
o Is the research design appropriate?
o Are the methods adequately described?
ANSWER
We have checked the manuscript and corrected all spelling errors
We thank the reviewer for his/her comments

Reviewer 2 Report
I have carefully revived the manuscript entitled "Combined magnetic hyperthermia and photothermia with magnetic nanorods". The manuscript is well written and the content is of scientific significance. I would suggest the authors to modify based on the below comments which I can recommend publication.
1. The abstract is very extended and needs to be shortened.
2. In the introduction section, the use of autofluorescence can be done for tracking and understanding the mechanism more precisely. I would suggest authors to add this discussion in the introduction section using the below references.
Kumar, P. et al. Low-temperature large-scale a. hydrothermal synthesis of optically active PEG-200 capped single domain MnFe2O4 nanoparticles. Journal of Alloys and Compounds 904, 163992, (2022).
Kumar, P. et al. Observation of intrinsic fluorescence in cobalt ferrite magnetic nanoparticles by Mn2+ substitution and tuning the spin dynamics by cation distribution. Journal of Materials Chemistry C 10, 12652-12679, (2022).
3. I would suggest moving some of the figures into supporting figures as it will help in improving the originality and novelty.
Author Response
Reviewer: 2
I have carefully reviewed the manuscript entitled "Combined magnetic hyperthermia and photothermia with magnetic nanorods". The manuscript is well written, and the content is of scientific significance. I would suggest the authors to modify based on the below comments which I can recommend publication.
COMMENT 1
The abstract is very extended and needs to be shortened.
ANSWER
The abstract has been rewritten and shortened as suggested. For the reviewer’s convenience we reproduce below the new text:
Abstract: Magnetite nanorods (MNRs) are synthesized based on the use of hematite nanoparticles of the desired geometry and dimensions as templates. The nanorods are shown to be highly monodisperse, with a 5:1 axial ratio, and with 275 nm long semiaxis. The MNRs are intended to be employed as magnetic hyperthermia and photothermia agents, and as drug vehicles. To achieve a better control of their photothermia response, the particles are coated with a layer of gold. Magnetic hyperthermia is performed by application of alternating magnetic fields with frequencies in the range 118-210 kHz and amplitudes up to 22 kA/m. Photothermia is performed by subjecting the particles to a near-infrared (850 nm) laser, and three monochromatic lasers in the visible with wavelengths 480 nm, 505 nm, and 638 nm. Best results are obtained with the 505 nm laser, because of the proximity between this wavelength and that of the plasmon resonance. A so-called dual therapy is also tested, and the heating of the samples is found to be faster than with either method separately, so that the strengths of the individual fields can be reduced. Finally, the release of the antitumor drug doxorubicin is investigated for the first time in the presence of the two external fields, and of their combination, with a clear improvement in the rate of drug release in the latter case.
COMMENT 2
In the introduction section, the use of autofluorescence can be done for tracking and understanding the mechanism more precisely. I would suggest authors to add this discussion in the introduction section using the below references.
Kumar, P. et al. Low-temperature large-scale a. hydrothermal synthesis of optically active PEG-200 capped single domain MnFe2O4 nanoparticles. Journal of Alloys and Compounds 904, 163992, (2022).
Kumar, P. et al. Observation of intrinsic fluorescence in cobalt ferrite magnetic nanoparticles by Mn2+ substitution and tuning the spin dynamics by cation distribution. Journal of Materials Chemistry C 10, 12652-12679, (2022).
ANSWER
A new paragraph has been included in page 3 of the revised version regarding the possibilities of the use autoluminescent particles, as follows
Although magnetite NPs by themselves have a photothermal response to a λ ≈ 800 nm laser excitation, as demonstrated by Espinosa et al. [20] and other authors [8], pursuing multi-functional uses of these nanosystems, the combination of MNPs with metallic surfaces, especially gold coatings, has gained importance in recent years. This is due to the possibility of adding an extra optical response to the magnetic one and hence, maximizing the power absorption when exposed to an electromagnetic field.
It is worth mentioning that it has recently been shown that some ferrites may show photoluminescence behaviour: the substitution of Fe2+ by Mn2+ in magnetite or in cobalt ferrite gives rise to such optical activity [21,22]. Interestingly, this may open an additional functionality to the MNPs as sensors and active devices. Thus, Ortgies et al. [23] make use of hybrid structures (MNPs and infrared emitting PbS quantum dots) to track, and thus observe deep tissue images with higher penetration by magnetic resonance and luminescence. However, the role of such structural changes in the photothermal response can also be a field of application, yet unexplored.
COMMENT 3
I would suggest moving some of the figures into supporting figures as it will help in improving the originality and novelty.
ANSWER
Following the reviewer’s suggestion, the following changes have been introduced:
- 1 and 2 have been merged into a single Figure 1
- Former Fig. 2 has been moved to Fig. S1
- Former Fig. 5 has been moved to Fig. S2
- Former Fig. 9 has been moved to Fig. S3
- Former Fig. 15 has been moved to Fig. S4
- Former Figure 17b has been moved to Figure S5
We thank the reviewer for his/her useful comments on our paper

Reviewer 3 Report
The paper presents experimental analysis of hyperthermal and photothermal behavior of the developed nanoparticles with magnetic nanorods. The research performed is interesting and is worth of being published after some necessary corrections. However surprisingly the authors have chosen the wrong journal for their paper. There are several journals which suit much better for the subject matter of the paper: sophisticated magnetic nanoparticles aimed for cancer treatment.
I would recommend to transfer the paper to one of the following MDPI journals: Magnetochemistry, Nanomaterials, Pharmaceuticals, Materials or Biomedicines. Or to Advanced Magnetic and Optical Materials, Biochemical Engineering Journal, Journal of Magnetism and Magnetic Materials etc.
Author Response
Reviewer: 3
COMMENT
The paper presents experimental analysis of hyperthermal and photothermal behavior of the developed nanoparticles with magnetic nanorods. The research performed is interesting and is worth of being published after some necessary corrections. However surprisingly the authors have chosen the wrong journal for their paper. There are several journals which suit much better for the subject matter of the paper: sophisticated magnetic nanoparticles aimed for cancer treatment.
I would recommend to transfer the paper to one of the following MDPI journals: Magnetochemistry, Nanomaterials, Pharmaceuticals, Materials or Biomedicines. Or to Advanced Magnetic and Optical Materials, Biochemical Engineering Journal, Journal of Magnetism and Magnetic Materials etc.
ANSWER
Thanks for the comment and suggestion. We prefer Polymers as the main journal, since the importance of the coated layer affects the behaviour of the magnetic characteristics, and thus the hyperthermia response. In addition, the PEI coating provides biocompatibility to the nanoparticles, prevents large aggregates from forming and the triple layer confers stability to the system over time. All these factors are key to its application in future "in vitro" and "in vivo" experiments.

Reviewer 4 Report
The paper describes dual magneto-optically active systems with a potentially bioapplications. The work is properly designed and performed, however, several points should be adressed.
1. Writing style and grammar should corrected. There are several alternations of present and past tenses in consequent sentences in the text. Some typo occurs (for instance PEI/PSS/Pei in Conclusion)
2. All figure and some literature references are broken.
3. Figure 4. Section a should stand on the left and и on the right
4. Figure 6b clearly demonstrates that the synthesized material is a mixture of alpha- and gamma- maghemites that is evidenced by bands at 283, 379 and 502 cm. (see doi:10.2147/IJN.S256542; 10.1366/000370209788559539; 10.1002/(sici)1097-4555(199711)28:11<873::aid-jrs177>3.0.co;2-b).
Author Response
Reviewer: 4
The paper describes dual magneto-optically active systems with a potentially bioapplications. The work is properly designed and performed; however, several points should be adressed.
COMMENT 1
Writing style and grammar should corrected. There are several alternations of present and past tenses in consequent sentences in the text. Some typo occurs (for instance PEI/PSS/Pei in Conclusion)
ANSWER
Thank you for your comment, the article has been corrected as suggested.
COMMENT 2
All figure and some literature references are broken.
ANSWER
This is because this is plain text. The final and eventually published version will keep links active.
COMMENT 3
Figure 4. Section a should stand on the left and и on the right
ANSWER
Thank you for your comment, this error has been corrected.
COMMENT 4
Figure 6b clearly demonstrates that the synthesized material is a mixture of alpha- and gamma- maghemites that is evidenced by bands at 283, 379 and 502 cm. (see doi:10.2147/IJN.S256542; 10.1366/000370209788559539; 10.1002/(sici)1097-4555(199711)28:11<873::aid-jrs177>3.0.co;2-b).
ANSWER
The reviewer is right. Our particles are a mixture of magnetite and maghemite. This has been mentioned in the new version (page 8) as follows:
It can be seen in Figure 3a that a complete transformation of hematite into magnetite has been achieved in view of the XRD results (magnetite reference data taken from RRUFF database [33]). However, since the XRD diffraction pattern of magnetite and maghemite are very similar, other methods can be used to ascertain the structure. In addition to XPS, Raman spectroscopy is one possibility. The Micro-Raman spectroscopy results obtained with our sample of MNRs are plotted in Figure 3b: the bands observed correspond to maghemite (288, 406, and 495 cm-1 reference data, RRuff ID R140712 [33]), and magnetite (309, 560, and 690 cm-1, RRuff ID R060656.3 [33]), indicating that our particles are in fact a mixture of maghemite and magnetite, although the magnetite denomination will be used throughout the manuscript for simplicity.
We thank the reviewer for his/her analysis of our manuscript.

Round 2
Reviewer 1 Report
I have suggested many points to the author for improving the quality of the paper, but I found no improvement in the current version. Therefore, I believe that the present paper is not novel, and it must undergo extensive and significant revisions before it can be published.
Author Response
MANUSCRIPT
Polymers-1906948
Combined magnetic hyperthermia and photothermia with magnetic nanorods
Response to Reviewers Comments (2nd. round)
(Italics: original comments. Regular: answers)
Reviewer: 1
COMMENT 1
I have suggested many points to the author for improving the quality of the paper, but I found no improvement in the current version. Therefore, I believe that the present paper is not novel, and it must undergo extensive and significant revisions before it can be published.
ANSWER
We regret that we did not receive the many suggestions mentioned by the reviewer in the first round. Perhaps due to some mistake, the only comment that reached us (and which was responded in the first round of answers) was the following one:
We will be glad to consider the additional suggestions raised by the reviewer.
Note however, that the manuscript has extensive change accordingly to reviewer´s comments
We thank the reviewer for his/her comments

Reviewer 3 Report
In the corrected version of the manuscript I still didn’t find any clarifying info supporting the importance of the polymeric layer. If authors consider this layer so important for the targeted performance of their nanoparticles it should be indicated in the title, abstract and keywords. All these items are completely free of even mentions of this layer…
An extended discussion on advantages of this layer along with corresponding references to works of other authors should be also added to the Introduction section.
The very important aspects are the particles toxicity and details of their excretion from the body. These aspects are beyond the experimental part of the paper, however they should be addressed in the Introduction on the specific examples of similar particles examined by other authors. The only one given reference [24] accompanied with vague words is insufficient. Authors should make a detailed search and provide a comprehensive info on this subject.
Author Response
MANUSCRIPT
Polymers-1906948
Combined magnetic hyperthermia and photothermia with magnetic nanorods
Response to Reviewers Comments (2nd. round)
(Italics: original comments. Regular: answers)
Reviewer: 3
COMMENT 1
In the corrected version of the manuscript I still didn’t find any clarifying info supporting the importance of the polymeric layer. If authors consider this layer so important for the targeted performance of their nanoparticles it should be indicated in the title, abstract and keywords. All these items are completely free of even mentions of this layer…
An extended discussion on advantages of this layer along with corresponding references to works of other authors should be also added to the Introduction section.
ANSWER
We apologize for not correctly interpreting the reviewer’s previous questions. A thorough discussion has been added in the Introduction section, and the Abstract has also been modified to account for these facts. Finally, polyelectrolyte layer and polyethyleneimine have been added to the keywords.
The title is now:
Combined magnetic hyperthermia and photothermia with polyelectrolyte/gold-coated magnetic nanorods
The following sentence has been added to the Abstract:
Because of toxicity concerns with PEI coatings, viability of human hepatoblastoma HepG2 cells was tested after contact with nanorods suspensions up to 500 µg/mL concentration. It was found that the cell viability was indistinguishable from control systems, so the particles can be considered non-cytotoxic in vitro.
The following paragraphs have been added to the Introduction (pages 3,4):
The attachment of gold nanoparticles to magnetite has been explored by different authors, and the stability of the coating and its uniformity and thickness are always an issue to consider. In the present approach, the core particles were coated with a cationic polyelectrolyte, poly(ethylene imine) (PEI) in order to promote the adhesion of negatively charged gold nanoparticles, as well as increasing the stability of the composite particles against aggregation [37,38]. This improved stability has been taken advantage of in magnetic hyperthermia, using different stabilizing routes [39,40].
The use of this polymer in the biomedical field has been strengthened by its application in gene delivery as a therapy in the treatment of different diseases [41], as an alternative to viral vectors, presently showing the best performance in this task [42,43]. Due to its ability to electrostatically binding DNA, PEI has become one of the most widely used alternatives to viruses for the transport of genetic material into cells. There are, however, concerns about the cytotoxicity of PEI [41,44,45]. For instance, Hu et al. [46] found that cardiovascular toxicity in zebra fish embryos was associated to branched PEI with a molecular weight of 25 kD, and that toxicity increased with molecular weight. Although this is the most usual vehicle of DNA for gene transfection, lower molecular weights appear as less toxic to cells, with a safe limit around 2 kDa [47,48], at the price of reducing transfection efficiency.
Interestingly, Wang et al. [49] found that building of a layer-by-layer assembly of citric acid and PEI reduced significantly the cytotoxicity of the polymer. This is an argument in favor of coating the gold particles attached to the magnetite cores investigated in this work with citrate anions to make them negatively charged. Another advantage of attaching gold to the particles was demonstrated by Arsianti et al. [21], who found that the cellular viability in the presence of magnetite/gold/PEI used as DNA vehicles was significantly higher than that measured in the absence of the gold layer.
COMMENT 2
The very important aspects are the particles toxicity and details of their excretion from the body. These aspects are beyond the experimental part of the paper, however they should be addressed in the Introduction on the specific examples of similar particles examined by other authors. The only one given reference [24] accompanied with vague words is insufficient. Authors should make a detailed search and provide a comprehensive info on this subject.
ANSWER
The general aspects of the toxicity of particles similar to ours are detailed in the following points (pages 2-3):
The most frequently used materials are magnetite (Fe3O4) and maghemite (γ− Fe2O3) since they are magnetic particles accepted for clinical use in Europe and USA. This is because of their strong magnetization and minimal toxicity, the latter being greatly improved by proper surface treatments of the particles [19]. In fact, although the magnetic core plays a fundamental role in the pursued applications, the easy aggregation and degradation of iron oxides in biological fluids prevents the use of the bare particles in biomedical uses. Ideally, particles should have diameters below 6 nm in order to be excreted through the kidney [20]. However, few syntheses produce such small particles, and their use by injection and transport by an externally applied magnet would be difficult due to their Brownian random motions. Whatever the size, untreated particles are prone to adsorb plasma proteins (opsonins) that make them targets for macrophages of the mononuclear phagocyte system, leading to changes in the particles themselves and in the pH of the medium, affecting cell viability [21-23]. Coating with inert or biocompatible shells appears necessary for reducing the potential toxicity of the particles, including polymers, silica, or inert metals as gold or silver [20,23-27].
Many tests of cell toxicity are performed in vivo, clearly simpler, and more accessible than in vivo evaluations, but this brings about the issue of the correspondence between both types of results, with poor correlation found in most cases. Mahmoudi et al. [22] analyzed the possible routes to increase the applicability of in vitro toxicity evaluations as predictor of in vivo results. For instance, Maniglio et al. [28] recently demonstrated that magnetite/gold composite nanoparticles 12 nm in diameter did not show any significant toxicity against MG63 and NIH/3T3 cell lines, confirming results from other authors [29,30] in relation to the negligible cytotoxicity of gold-coated magnetite. A one year lasting study performed by Kolojsnaj-Tabi and coworkers [31] disclosed many aspects of the fate of gold/magnetite nanocomposites in vivo, after injection in laboratory mice. It is worth noting that the degradation of magnetite may enter the route of iron metabolism by cells and be finally managed by the organism of the living animal without provoking further harm, although at the price of losing the original magnetic properties. These authors found that particles accumulated in lysosomes of Kupffer cells in liver and macrophages in spleen. After long-time observations, gold particles were found to remain rather stable (with some reduction in diameter), whereas only traces of iron oxide were appreciable. This was hence eliminated by dissolution, and not as solid nanoparticles.
In addition, we have carried out experiments on the toxicity of our magnetite particles. The experimental methodology is detailed on page 6, as follows:
2.2.8 Cytotoxicity determinations of PEI/PSS/PEI coated MNRs
The human hepatoblastoma cell line HepG2, supplied by European Collection of Animal Cell Cultures (Salisbury, U.K.) was used to perform cell viability experiments in order to prove the negligible toxicity of the triple polymer layer-coated MNPs. The cells were seeded into 96-well plates (10 000 cells/well) and cultured in a cell culture medium for 24 h with different concentrations of polymer-coated MNRs (25, 50, 100, 200, 300, 500 μg/mL) in a CO2 (5%) atmosphere. After the treatment was completed, the medium was removed from each well by aspiration, and the cells were fixed with 100 μL of glutaraldehyde. The plate was stirred for 15 min at 50 rpm and it was then washed with distilled water 8 times. After drying, 200 µL of 0.1% crystal violet was added and stirred for 20 min at 30 rpm. Again, the plate was washed and dried. Finally, the dye was solubilized with 100 μL of 10 % acetic acid. The plates were stirred for 10 min and introduced into a plate reader set at 590 nm wavelength, which provided the results of live cells [52].
And the corresponding results of the experimental assessment of cytotoxicity are given on pages 11-12, as detailed below:
3.6 Cell viability experiments
Cell viability experiments were performed to determine whether the coatings provided the necessary biocompatibility of the systems. For this purpose, HepG2 cells were seeded for 24 hours in a CO2 oven with different concentrations (25, 50, 100, 200, 300, 400, 500 μg/mL) of PEI-PSS-PEI coated nanoparticles. The percentage of viability has been normalized with respect to the control cells. The experiments were conducted in triplicate. Thus, the results represent the mean ± SD of these experiments. As can be seen in Figure 6, the preliminary data are very promising. The cytotoxicity of polyelectrolyte-coated MNRs particles is negligible, even at the highest concentrations. No significant differences were observed between cells cultured with the concentration sweep and control cells (within statistical error).
Figure 6. Effects of PEI-PSS-PEI coated MNRs on HepG2 cells proliferation. The concentration of nanorods as indicated.
We thank the reviewer for his/her comments

Round 3
Reviewer 1 Report
My old comments have been sent to the editor again, and they will forward them to you. I would appreciate it if you would address all of the comments and suggestions and improve the manuscript accordingly.
Author Response
Dear colleague
Attached you can find our answers to your comments on our manuscript. Once again, we apologize for our late answer, but due to a technical problem your original report did not reach us on time.

Reviewer 3 Report
The authors addressed all my comments and I can now recommend publication of the paper in the current form.
Author Response
Thank you very much for your analysis of our manuscript
Round 4
Reviewer 1 Report
.